# A Mystic Weed, *Parthenium hysterophorus*: Threats, Potentials and Management

H. M. Khairul Bashar [1,2], Abdul Shukor Juraimi [1,*], Muhammad Saiful Ahmad-Hamdani [1], Md Kamal Uddin [3], Norhayu Asib [4], Md. Parvez Anwar [5] and Ferdoushi Rahaman [1]

1 Department of Crop Science, Faculty of Agriculture, University Putra Malaysia (UPM), Serdang 43400, Selangor, Malaysia; gs53632@student.upm.edu.my (H.M.K.B.); s_ahmad@upm.edu.my (M.S.A.-H.); gs53678@student.upm.edu.my (F.R.)
2 Bangladesh Agricultural Research Institute (BARI), Gazipur 1701, Bangladesh
3 Department of Land Management, University Putra Malaysia (UPM), Serdang 43400, Selangor, Malaysia; mkuddin@upm.edu.my
4 Department of Plant Protection, Faculty of Agriculture, University Putra Malaysia (UPM), Serdang 43400, Selangor, Malaysia; norhayuasib@upm.edu.my
5 Agro Innovation Laboratory, Department of Agronomy, Faculty of Agriculture, Bangladesh Agricultural University, Mymensingh 2202, Bangladesh; parvezanwar@bau.edu.bd
* Correspondence: ashukor@upm.edu.my; Tel.: +60-397-694-940

**Abstract:** *Parthenium hysterophorus* is an invasive weed species that competes aggressively with other plants and is also allelopathic. It poses a significant risk to human health, livestock, the environment, soil, and agriculture. However, given some clinical studies, its potential for antidiabetic, antioxidant, antitumor, herbicidal, pesticidal, and antimalarial therapies should be researched further in attempts to discover more relevant applications. It can be used as a nutrient-dense, readily available, and cheap fertilizer. Parthenium can also be used as an herbicide, an insecticide, and a phyto-remedial mediator to extract metals and dyes from agricultural waste. Here we provide basic information on the morphology, reproduction, environmental impacts, and management of this species. Effects of methanol, ethanol, hexane, acetone, and aqueous (water) Parthenium extracts are described. Because *P. hysterophorus* is said to be one of the world's seven worst weeds, some control measures, including mechanical, chemical, cultural, and biological control, are discussed. The allelopathy of this weed is difficult to regulate, and there are both positive and negative interactions between Parthenium and other species due to allelochemical action. Several toxic phenolic compounds produced by *P. hysterophorus* are responsible for weed suppression, and we discuss details of their mode of action and potential applications.

**Keywords:** Parthenium; allelopathy; allelochemicals; herbicidal effects; suppression; seedling growth

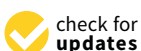



## 1. Introduction

Parthenium (*Parthenium hysterophorus* L.) is a weed that produces many seeds and has spread widely throughout Asia and other areas beyond its native range in Central and South America and the southern USA. Post-1955, Parthenium has become a major problem as a weed in agro-habitats in over 30 nations [1]. *P. hysterophorus* infests a variety of landscapes, including farmsteads, fallow ground, orchards, and railway tracks [2]. Parthenium weed is a member of the Asteraceae family, a large and diversified species with a worldwide distribution [3]. Due to its many features, such as its short life cycle of 90–120 days, adaptability to photo-thermal conditions, a lack of natural enemies (usually), and rapid growth, ability to spread via waterways and roadways [4], it is able to spread very quickly.

It is suspected that Parthenium was spread by vehicles or as a contaminant of seed lots in many cases [5]. The spread of its seeds depends on water supply, livestock, and machinery movement. It has other common names, such as Santa Maria feverfew; bitter

weed, which refers to the herbal flowering plant; and (sometimes) carrot grass. It is a fast-maturing annual weed and with a deep taproot and an erect stem that becomes woody with age. Due to its allelopathic influence, this weed is thought to cause allergic respiratory problems, mutagenicity in humans and livestock, and severe reductions in crop production [6]. According to Bajwa et al. [7], this toxic, aggressive species is one of the worst weeds worldwide. This weed is responsible for multi-million-dollar losses in Australia and is considered a harmful crop in more than 45 countries [8,9]. This plant is not only harmful to agriculture but also is a major factor in multiple human diseases. Among these are asthma, cancer, allergies, and stomach diseases [10]. On the other hand, its potential benefits have been identified by various researchers. It is used as an antioxidant, an anticancer agent, and an antitumor agent, and its extracts are often used as pesticides to control diseases [10]. It is often used as an organic fertilizer because the proportions of N, Mg, Ca, K, and other nutrients are high in this plant, so it provides value to plant growth [11]. Parthenium improves seed germination, seedling growth, biomass, and yield indices for certain crops [12]. Parthenin, hysterin, hymenin, and ambrosin comprise important compounds found in Parthenium plants [10].

Some of the most significant chemical compounds in Parthenium plants exhibit allelopathic activity. Allelopathy is a mechanism through which phytochemicals are produced by one plant, which increases or decreases the germination rate of another plant [13]. Allelopathy describes the beneficial or harmful effects of one plant on another plant, e.g., crop and weed species interact allelopathically. Allelopathy is caused by the release of biochemicals known as allelochemicals from plant parts through leaching, root exudation, volatilization, residue decomposition, and other processes in both natural and agricultural systems [14]. Allelopathic plants emit chemical compounds into the soil from their roots, and as nearby plants absorb these chemicals, they are inhibited or even destroyed. While an allelopathic effect is generally considered to be harmful, there may also be some beneficial effects, depending on the allelochemical target and other factors [15]. Allelochemicals produce cytotoxicity through physiological effects [16]. Allelochemicals cause a variety of physiological effects, such as water absorption, leaf area expansion, and mineral nutrient absorption, which all decrease plant growth. Various allelochemicals were shown to reduce the absorption of indole-3-acetic acid (IAA) oxidase and other important macronutrients and micronutrients in germinating mustard seedlings by inhibiting root dry weight and root moisture content [16].

The first major objective of this study is to provide background information on the Parthenium weed, including its characteristics, reproduction, and benefits and drawbacks. Our second major objective is to review the effects of *P. hysterophorus* on human health and how to treat them, its herbicidal effects using various solvents, allelopathic effects, the mechanisms of action involved in allelopathy, and the economic benefits of Parthenium.

## 2. A Botanical Description of Parthenium and Its Distribution

*P. hysterophorus* is an annual herbaceous plant that reproduces mostly through seeds (Figure 1). Following sprouting, the young plant has a basal rosette of bright green and finely lobed leaves that measure about 8–20 cm in length and 4–8 cm in breadth. During unfavorable conditions, the rosette stage can continue to grow up to a maximum of 2.5 m long [3,17]. Both leaves and stems have short and fluffy hair or trichomes, four styles of which have been recognized and considered for their taxonomic significance [18]. The flower heads are terminal and somewhat hairy; they consist of several small white capitula-shaped florets. Usually, each head has five productive ray florets, although occasionally six or eight. Thousands of branches, which develop in separate clusters, produce compressed black seeds about 2 mm in size. Originally, Parthenium was found in the Gulf of Mexico, the USA, the West Indies, and Central America [19]. Parthenium has now invaded 46 countries and regions [20]. Both Parthenium's life cycle and how it spreads are shown in Figure 2.

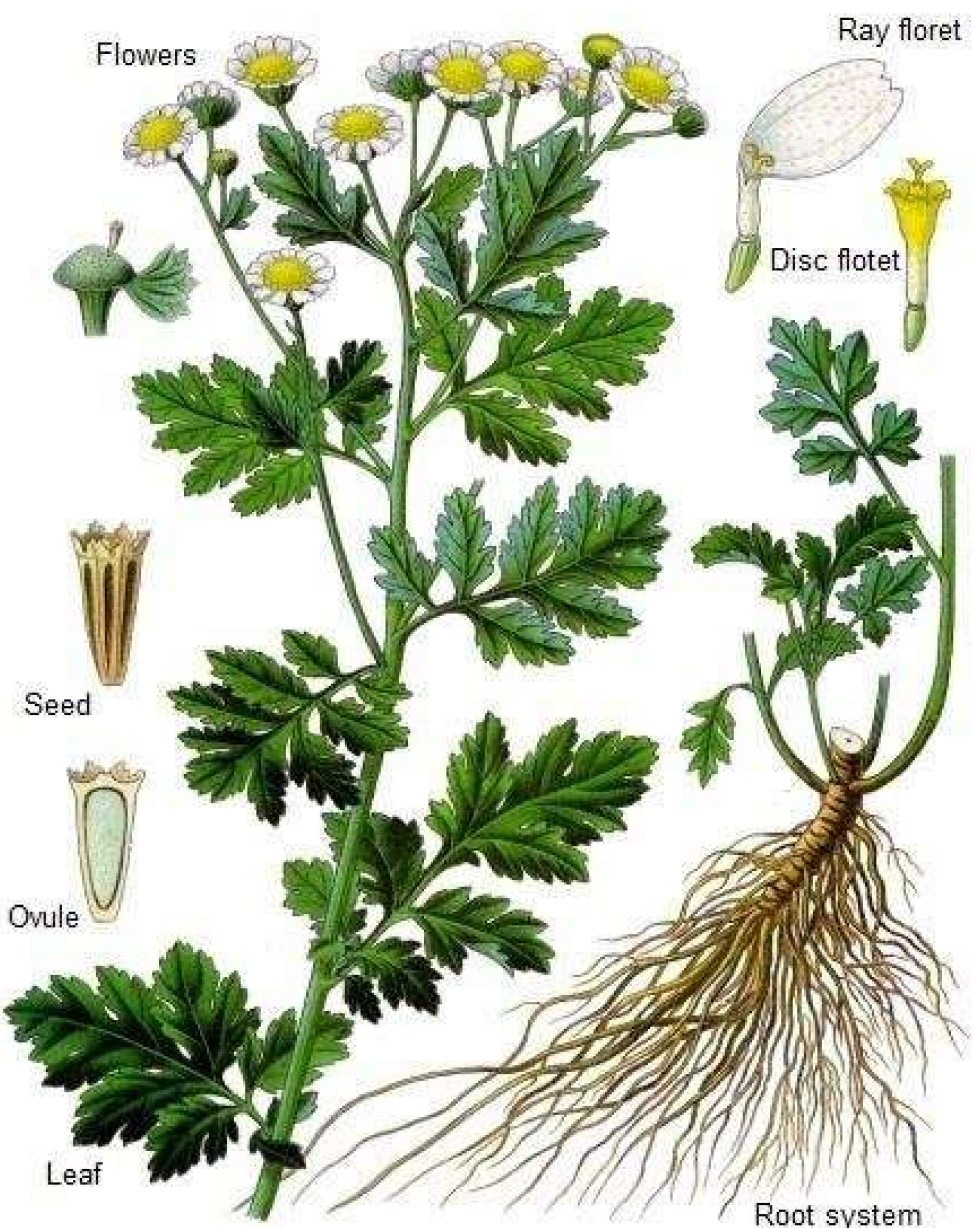

**Figure 1.** The different plant parts of *Parthenium hysterophorus* [21].

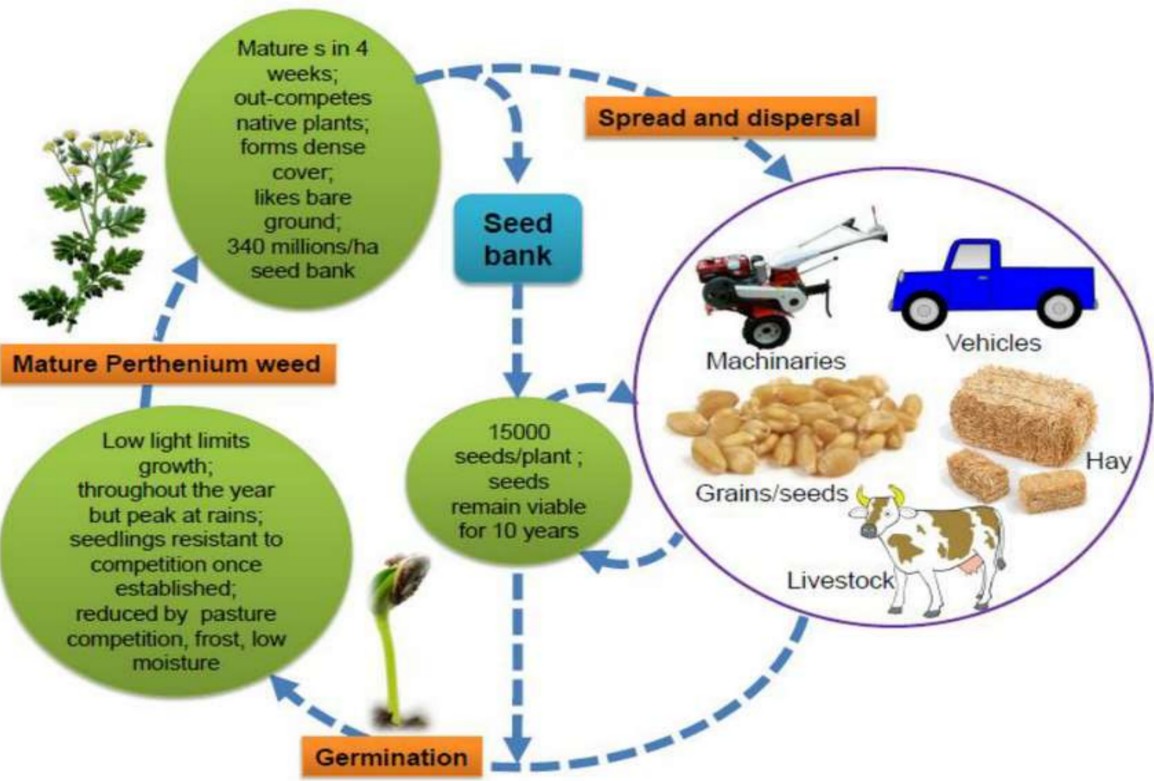

**Figure 2.** The life cycle of Parthenium weed [21].

### 3. The Harmful Effects of Parthenium

*3.1. Human Health*

Parthenium roots can cause allergic diseases, such as photodermatitis, asthma, hay fever, skin rashes, excessive water loss, peeling skin, swelling, and itching of the mouth and nose [22]. Some of the main elements contained in Parthenium are chlorogenic acid, anisic acid, p-anisic acid, caffeic acid, and benzoic acids, which are very harmful to humans and livestock [23]. Hand weeding in Parthenium-infested fields can cause skin diseases, and Parthenium-related allergies can also cause malarial infection-related fever. Skin infections, allergies, eczema, fever, allergic rhinitis, dark spots, burning, and swelling around the eyes are all signs of long-term exposure to this herb. Diarrhea, extreme papular erythematous eruptions, and shortness of breath are all symptoms of *P. hysterophorus* [10]. Respiratory symptoms normally start with increased fever and respiratory problems and become more severe after 3–5 years of incremental exposure, resulting in asthma and allergic bronchitis [24]. A Parthenium plant in the rosette stage is shown in Figure 3. Its detrimental effects on crops, livestock, and the human body, alongside control methods, are shown in Figure 4.

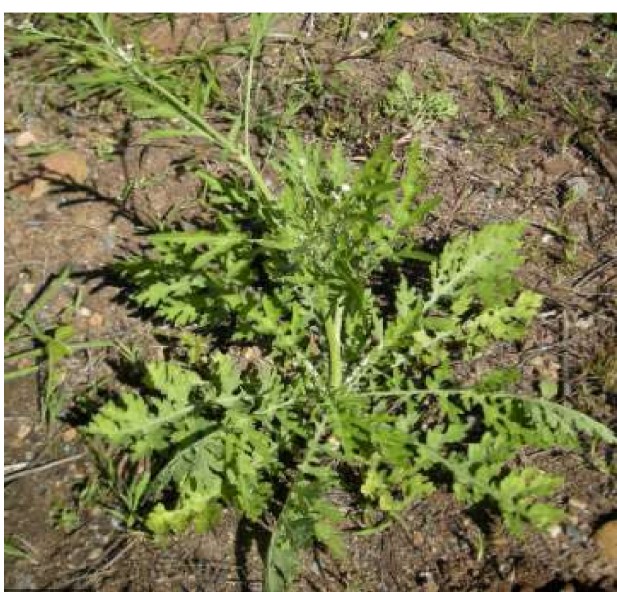

**Figure 3.** The rosette stage of Parthenium.

*3.2. Effects on Livestock*

Parthenium growing in grasslands can be virtually invisible, making it more likely that livestock will consume it, especially if grassland managers fail to manage it. Parthenium decreases livestock productivity by reducing the amount of forage. It may also affect grazing animals' welfare, milk production, and meat quality [24]. *P. hysterophorus*-fed buffalo and hybrid calves develop atrophic eruptions, alopecia, skin depigmentation, and anemia. In mature livestock, continuous feeding of *P. hysterophorus* for up to 12 weeks can cause anorexia and dermatitis (*Osmanabadi*) [2]. When cattle consume Parthenium or they come into contact with the weed on a regular basis, poisoning may result. Death, rashes on the body and udders, alopecia, loss of skin pigmentation, allergic skin reactions, dermatitis, diarrhea, anorexia, and pruritus are all possible outcomes for those animals. The psychological behavior of animals can also be influenced by Parthenium [25]. Parthenium silage has nutritional value that is fairly similar to that of a sheep's normal dietary requirements, and the seeds of Parthenium obtained from the silage did not germinate [26].

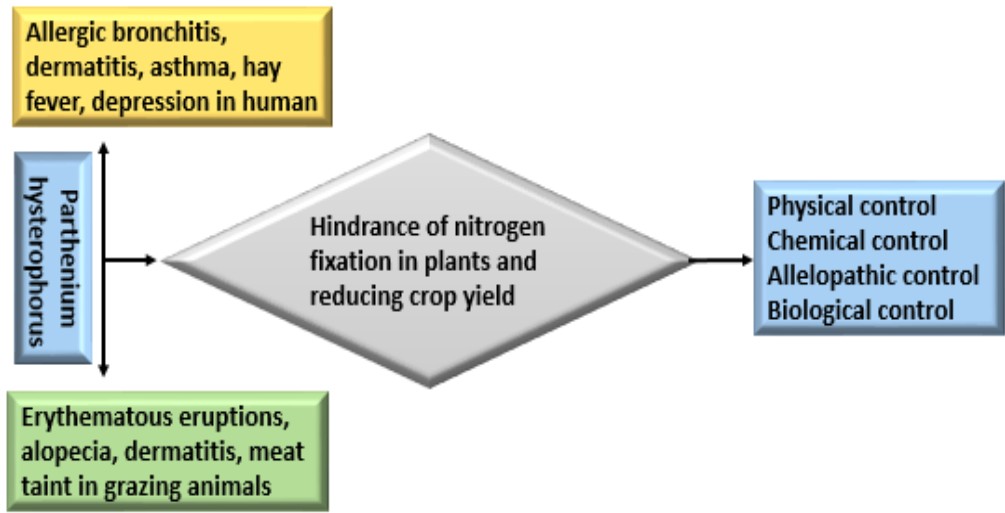

**Figure 4.** General detrimental effects of *Parthenium hysterophorus* and its control management [27].

### 3.3. Effects on Soil

With allelopathic plants present, there are strong correlations between crop development and soil microorganisms. Direct integration of Parthenium residues into soil has the potential to have allelopathic impacts on crop development and establishment [28]. As a result of *P. hysterophorus* expansion, native and non-native species have been hampered or replaced. Soil nitrogen and organic content were found to be considerably higher in infested plots than in non-invaded plots. The invaded plots had the highest levels of pH, phosphorus, and potassium; the non-invaded plots had low to moderate levels [29]. Parthenium weed can take nutrients even from nutrient-depleted soils, resulting in high tissue levels of nitrogen (3%), phosphorus (2%), potassium (4%), and other macronutrients and micronutrients, making it an excellent green manure for field crops [30]. According to Dogra and Shood [31], allelochemicals are formed by *P. hysterophorus* leaves and discharged into the soil by leaching or decomposition, and they affect plant growth directly or indirectly by affecting the soil's physico-chemical characteristics. According to Javaid and Shah [32], a 4% Parthenium treatment increased maize's biomass output to the same level as prescribed NPK fertilizers.

### 3.4. Effects on Crops

Several scholars have noted that Parthenium can have positive or negative effects on various crops [33]. In East Africa, sorghum grain production was reduced by 40–97% if Parthenium was allowed to grow unchecked for the entire season [34]. Parthenium, on the other hand, had an allelopathic impact on the sprouting and the development of other crops [22]. Germination period, growth rate, and yield reductions of Indian traditional crops resulted from soils being infected by the Parthenium weeds because the leaves contain p-coumaric acid and caffeic acid. These are responsible for growth inhibition [35]. Seedling growth of *Zea mays* was seriously disturbed in another study, and crucifers' seedling elongation was affected by Parthenium leaf extracts [36]. *P. hysterophorus* has an influence on bean growth at low concentrations, and Parthenium ash concentration levels can boost seed germination, radicle length, and biomass production [36]. When *P. hysterophorus* residues are combined in soil, they have a negative impact on the germination and subsequent seedling growth of native plants [31]. In a number of dicot and monocot plants, parthenin has been described as a germination and radical growth inhibitor [37]. The allelopathic characteristics of Parthenium make it difficult for agricultural crops such as wheat, rice, maize, pigeon pea, sorghum, and black gram to germinate and flourish, and their yields may decrease by as much as 40% [27].

### 3.5. Other Affected Areas and Animals

A huge number of national parks, tourism sites, and industrial sites may be affected by Parthenium, and wild animals may suffer the consequences. For example, most national botanical gardens in Africa [38], India [39], South Africa [40], and Ethiopia [41] have been affected by this weed.

## 4. The Beneficial Effects of *P. hysterophorus*

There are a considerable number of beneficial effects of Parthenium (Table 1).

### 4.1. Antidiabetic Effects

*P. hysterophorus* aqueous extract showed strong hypoglycemic action. Within 2 h, fasting blood glucose levels in alloxan-induced diabetic rats dropped significantly ($p < 0.01$) [2]. As a result, this treatment may be helpful, mainly for type II diabetics who are insulin-independent [42–44]. Fever, neurological conditions, diarrhea, urine infections, malaria, and emmenagogue have all been treated with distilled *P. hysterophorus* liquor in traditional medicine, and women's vaginal and urinary disorders have sometimes been treated using tea produced from the leaves and roots of *P. hysterophorus* [45]. In addition, some tribal

people use it to treat itching, skin conditions, rheumatic pain, eczema, heart complications, and reproductive problems.

**Table 1.** Beneficial effects of *Parthenium hysterophorus* weeds.

| Beneficial Effects | Features | References |
|---|---|---|
| **Health Benefits** Antidiabetic | Aqueous extract exhibited significant hypoglycemic activity | [42–44] |
| Antioxidant | Methanolic extracts showed the high antioxidant effect | [11,46,47] |
| Antitumor | Consumption of its flower resulted in a change in neoplastic markers, which slowed tumor development | [47] |
| Antimicrobial | Ethanolic extraction of *E. coli, A. niger, P. aeruginosa, Candida albicans,* and *Fusarium oxysporum* | [11,48] |
| Larvicidal | Larvae are attracted to root and stem extracts | [49,50] |
| Signal transduction | Blocker of the depolarizing neuromuscular junction | [51] |
| **Agriculture** Compost | Higher levels of nitrogen, potassium, and phosphorus, as well as a substantial amount of organic carbon, C/P, and the C/N ratio | [52–54] |
| Herbicide | *Euphorbia prostrata, Digitaria sanguinalis, Echinochloa crus-galli, Xanthium strumariam,* and *Portulaca oleracea* are weed candidates for control by Parthenium. | [55,56] |
| Pesticidal effect | Antifeedant action against *Spodoptera litura* Insecticidal activity against *Callosobruchus aculatus* Phytotoxic activity against *Cassia tora* Nematicidal activity against *Meloidogyne incognita* | [49,57] |
| **Waste Treatment** Heavy metal and dye removal | Wastewater treatment of nickel and methylene blue dye | [10,58,59] |

### 4.2. Antioxidant Activity

Due to their carcinogenic effects, free radicals are considered to be contributors to certain diseases. Synthetic antioxidants are regarded as contributing factors [46]. That is why natural antioxidants have drawn scientists' interest. When compared to Stevia's (*Stevia rebaudiana* Bertoni) effect, the methanolic extracts of *P. hysterophorus* demonstrated significant antioxidant activity [11]. As a result, this plant could be a suitable natural antioxidant source. A new, potent natural antioxidant could be made commercially available after investigating Parthenium for its active antioxidant ingredient(s) [2].

### 4.3. Antitumor Activity

Alberto Ramos [60] showed that *P. hysterophorus* extracts have antitumor potential through in vitro activity, with promising results in terms of tumor size reduction. The levels of neoplastic indicators, such as glutathione, cytochrome P-450, glutathione transferase, and UDP-glucuronyl transferase, were significantly altered, which slowed tumor growth and enhanced survival [2]. *P. hysterophorus* aqueous extract had hypoglycemic activity in alloxan-induced treated patients [43].

### 4.4. Antimicrobial Activity

Medically, Parthenium is known chiefly for its anticancer properties, but it may also be used for hepatic amoebiasis [61]. It has antibacterial, antifungal, and antiviral properties against *P. aeruginosa*, *E. coli*, and *Candida albicans*, respectively [48].

### 4.5. Larvicidal Effect

*P. hysterophorus* root and stem extracts are effective against mosquito larvae, particularly *Aedes aegypti* [50]. Chemical components extracted from leaves have significant effects on both the lifespan and the production of adult *Lipaphis erysimi* [49]. A new depolarizing

neuromuscular junctional block was observed on rats if Parthenium leaf extract was used as an alternative to anticholinesterase agents, such as neostigmine [51].

### 4.6. Parthenium Compost

*P. hysterophorus* is high in micronutrients and macronutrients, such as N, P, K, Ca, Mg, and chlorophyll, making it ideal for composting [62,63]. The plant's early growth, development, and dry matter output are all hampered by its high phenolic content. Hence, the composting of Parthenium and *Eichhornia crassipes* (a water weed rich in polyphenol oxidases) causes considerable reductions in phenol content, organic carbon content, and C/N and C/P ratios. Combining Parthenium and water hyacinth in compost provides a weed-control solution and a path to sustainable organic farming [2]. However, its phytotoxicity prevents plants from growing quickly and reduces dry matter harvests. Vermicomposting of Parthenium consumes nutrients and restricts unwanted plant noxiousness [53]. Furthermore, it also improves nutrient quality, which could be beneficial for organic farming and bioremediation [54].

### 4.7. Herbicidal Effects

Researchers have been focusing on plant-derived chemicals as eco-friendly herbicide alternatives for weed management for the past several decades. *P. hysterophorus* extracts reduced weed density significantly and also had allelopathic effects on *Eragrostis tef, Cynodon dactylon, Cyperus rotundus, Digitaria sanguinalis, Portulaca oleracea, Echinochloa crusgalli, Euphorbia prostrata, Xanthium strumariam,* etc. [2,56]. By suppressing cell division mediated by gibberellin and indole acetic acid, the sesquiterpene lactone is assumed to be responsible for its allelopathic influence on adjacent plants [62]. Kumar and Kumar [64] reported that ash concentrations above 3% reduced germination, seedling growth, development of plumule, and radicle length and biomass. There is substantial evidence that Parthenium extract could be used as a possible herbicide, given its effects on weed germination, density, and biomass. Hence, environmentally friendly, natural herbicides based on Parthenium could be derived as alternatives to synthetic herbicides [65].

### 4.8. Pesticidal Effects

The efficacy of Parthenium as an inoculum source was tested. *P. hysterophorus* is a serious pasture-invading weed that reduces pasture production 90% of the time [66]. It suppresses the growth of other plants in grasslands and pastures, thereby reducing fodder supplies [67].

### 4.9. Heavy Metal and Dye Removal

The degradation of the environment caused by heavy metals has become a global issue. Nickel (II) and cadmium (Cd) are used in silver factories, electroplating, zinc-based manufacturing, and Cd/Ni battery industries. Parthenium has shown the ability to absorb both nickel and methylene blue dye from wastewater and industrial waste. However, the highly toxic metals it absorbs can cause kidney disease, elevated blood pressure, bone deformity, and red blood cell (RBC) destruction. Ni and Cd can cause cancer and other diseases. Parthenium is a safe, affordable, and environmentally friendly absorbent of such industrial waste [2,10].

### 4.10. Other Economic Benefits

The capacity of *P. hysterophorus* to generate enzymes such as xylanase has been investigated [31,68]. Parthenium is very effective at removing dyes, dissolved heavy metals, and other contaminants, such as phenolics, from the atmosphere. It has benefited the imaging, mechanical, electronic, drug delivery, and molecular diagnosis industries greatly [31]. Nanotechnology has emerged as a remarkable tool for addressing a variety of challenges in everyday life. It has proved extremely advantageous to the imaging, mechanical, electronic, medication delivery, and molecular diagnosis industries [69]. Researchers' interest in the

synthesis of NPs from *P. hysterophorus*, such as TiO2, AgNPs, and zinc oxide nanoparticles, has risen recently [70,71]. These nanoparticles can be antifungal, antibacterial, environmentally safe, and effective at preventing vector-borne diseases. Furthermore, when combined with manure, Parthenium has been used very effectively to produce biogas and in treating heavy metal pollutants through bioremediation [69]. Researchers have also found that nitrogen, calcium, magnesium, and potassium are present in high proportions in this plant. Hence, the application of organic manure produced with Parthenium could play a great role in production enhancement (Figure 5).

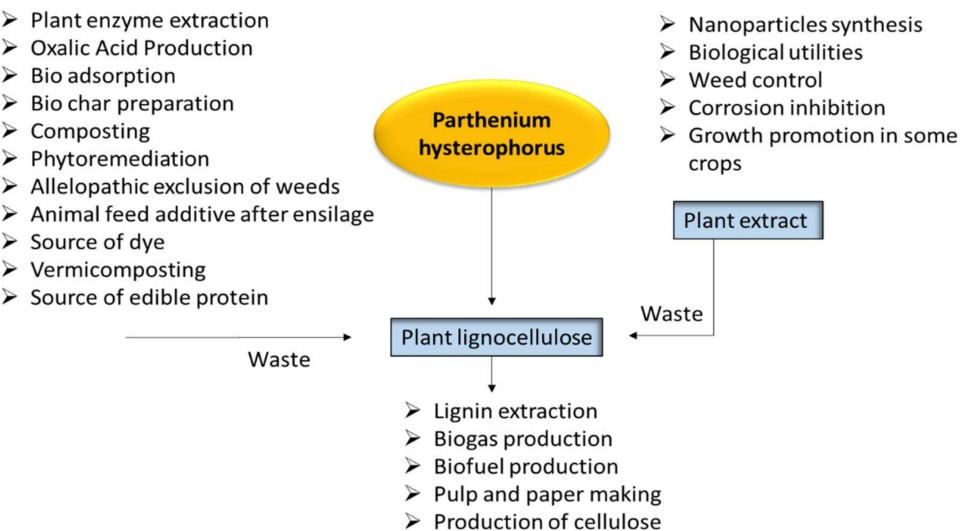

**Figure 5.** Sources of economic benefits of *Parthenium hysterophorus*.

## 5. Herbicidal Effects of Parthenium Extracts via Different Solvents

The herbicidal effects of the various parts of *P. hysterophorus* (leaf, stem, and flower) extracted via different solvents, such as methanol, acetone, hexane, ethanol, and water, are as follows.

### 5.1. Methanol Extracts

*P. hysterophorus* extracts have the inhibitory effect of reducing the early development of wheat, maize, and horse gram. Methanol extracts have more inhibitory power than aqueous extracts. This effect represents evidence of the allelopathic potential of *P. hysterophorus* root and shoots on agricultural production [72]. According to Dhawan and Gupta [73], methanol extraction is the optimal way to extract various active phytochemicals that cause inhibition. The germination method of *V. radiata* seeds was studied for up to 120 h using crude methanol extracts of Parthenium. The germination kinetics of crude extracts were studied, and it was discovered that there were substantial differences in germination kinetics among crude methanol treatments [74]. The methanolic extracts of Parthenium leaves and flowers had significant herbicidal effects, because allelochemicals are more abundant in the leaves and flowers than in the stems and roots [75]. Batish et al. [76] reported that the methanol extract and residues of *P. hysterophorus* inhibited the growth and development of various field crops. Methanol-isolated phytochemicals from the leaves, stems, and flowers of *P. hysterophorus* were able to affect weed and crop seedling germination and growth [77]. *P. hysterophorus* methanol extracts significantly reduced the radicle length of the target weed ($p = 0.05$) [65].

### 5.2. Ethanol Extracts

A Parthenium residue extract, which was rich in phenols, had robust phytotoxic effects on radish and chickpea growth [78]. The germination of *V. radiata* seeds was studied using ethyl acetate crude extracts of Parthenium for up to 120 h, and there were significant

differences in germination kinetics [74]. Evans [66] found the presence of a number of sesquiterpene lactones in the ethanol extracts of *P. hysterophorus,* including parthenin and coronopilin, and also reported the inhibition of seed germination and seedling growth for some crops. At greater concentrations, allelopathic chemicals present in ethanol root, stem, leaf, and flower extracts of Parthenium limited chickpea seed germination and seedling growth [75].

### 5.3. Hexane Extracts

Hexane-based Parthenium extracts had a major impact on the germination of all test organisms in the work of [74]. On wheat and barley seed germination, Parthenium hexane extracts had a marginal inhibitory effect [79]. A *P. hysterophorus* hexane extract caused a wide range of chromosomal abnormalities in dividing cells, which increased considerably with concentration and exposure time [79]. Arshad Javaid et al. [80] concurred that n-hexane extracts from rice cultivars are hazardous and affect crop germination and growth. When using n-hexane extracts of *A. nilotica* and *S. cumini*, there were reductions in germination, shoot length, root length, and fresh/dry biomass (30–35%, 20–27%, 50%, 50–55%, 80%, and 80–82%) for Parthenium [81]. When an n-hexane extracts of *P. hysterophorus* was administered, lettuce (*Lactuca sativa*) seed germination was entirely reduced [82]. The herbicidal activity of the *P. hysterophorus* n-hexane extracts was higher [83]. Singh et al. [84] discovered that a 60 g/L hexane extracts of onion bulbs reduced the cotton root and shoot heights. Of several hexane Parthenium extracts, the leaf extracts showed the most significant inhibitory effect; the others tested were stem and root extracts [85].

### 5.4. Acetone Extracts

According to Kohli et al. [86], acetone extracts of aerial portions of *P. hysterophorus* inhibited the sprouting and development of cabbage. The allelopathic potential varies from plant to plant [87], as was observed in horse gram, maize, and wheat. Wakjira et al. [88] also reported the inhibitory effect of *P. hysterophorus* on the development of seedlings on cereal crops and wild species. Parthenium acetone residues have inhibitory effects on the sprouting and growth of different plants, such as the radish and mustard trees. Zohaib et al. [89] found that the allelopathic effects of acetone extracts of several crop seeds resulted in considerable reductions in radical and plumule lengths for green grams. In cultivated colonies, a *P. hysterophorus* acetone extract provides a valuable zone of inhibition [90]. On the other hand, the phenolic compounds in the acetone extracts hindered lettuce and tomato germination at high doses [91].

### 5.5. Aqueous (Water) Extracts

According to Wakjira et al. [92], aqueous extracts of Parthenium have inhibitory effects on the early growth of wheat, maize, and horse gram. When seeds of *Zea mays* were treated with an aqueous extract of *P. hysterophorus* leaves, germination was reduced for all concentrations compared to the control, and the greatest inhibition was observed with 10% concentrations of *P. hysterophorus*. Leaf extracts of *P. hysterophorus* reduce the root and shoot elongation of *Oryza sativa*, wheat, maize, soybean plants, and some common Australian pasture grasses [93,94]. Five- and ten-percent stem and root extracts and 1% and 5% flower extracts affect shoot weight as well. Nevertheless, aqueous extracts of roots at only 1% promoted radicle growth in wheat and barley. The 10% extracts concentration was generally effective and even more powerful than the 1% and 5% extracts concentrations, specifically when the extracts was of the leaves. A similar effect of *P. hysterophorus* aqueous leaf extracts was also observed in cereals [79].

Tessema and Tura [79] found that aqueous extracts with concentrations between 8% and 10% had the highest inhibitory effects on *Zea mays* germination. Possibly, certain allelochemicals in the leaf extracts may have inhibited the embryo's growth or destroyed the embryo. The seed germination, plumule, and radicle length were reduced more as the concentration of aqueous extracts increased. Seed germination was reduced by increasing

the concentrations of aqueous extracts of Parthenium leaves and flowers, and sometimes complete germination failure was observed [95]. All water extracts reduced the germination percentage of *Artemisia dubia*. The effect was stronger when the concentration was raised, and the leaf extracts provided the most complete inhibition (stem and root extracts were also tested) [85]. The seed germination and seedling growth of lettuce were severely hampered by aqueous extracts of *P. hysterophorus* leaves and flowers [96]. According to Afridi and Khan [97], aqueous extracts of *P. hysterophorus* reduced seed germination, shoot length, fresh and dry biomass, and seed germination of *D. alba* when compared to the other treatments.

## 6. Controlling Parthenium

Stages of Parthenium control, management, and other actions are shown in Figures 6 and 7.

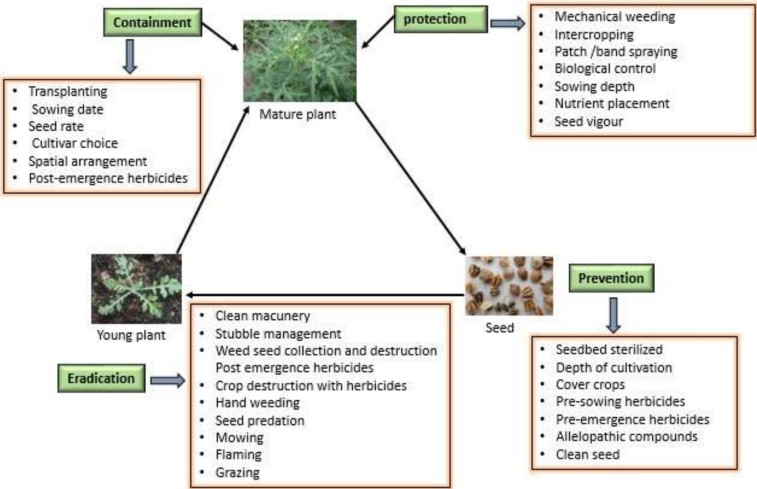

**Figure 6.** Management and control of Parthenium at different stages.

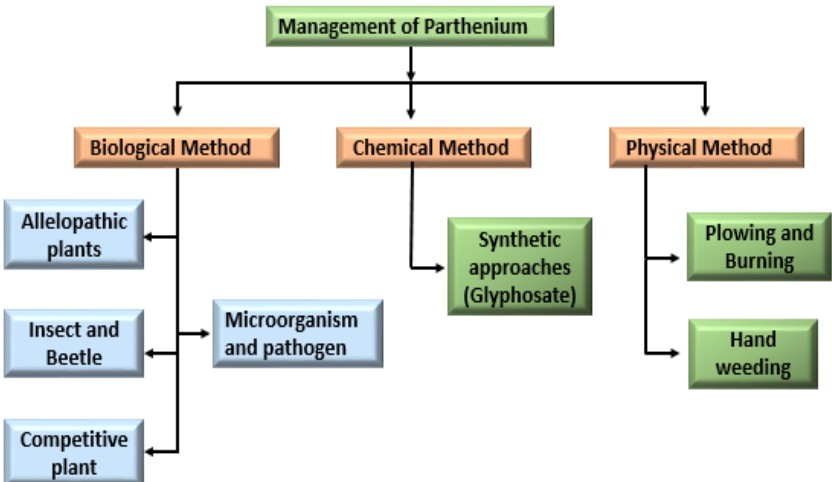

**Figure 7.** Schematic presentation of the management of *Parthenium hysterophorus* [27].

### 6.1. Use of Allelopathic Plants

Parthenium, in general, is a poisonous, pernicious, hazardous, allergenic, and aggressive weed that is a major hazard to humans, crops, and cattle. The use of plants with allelopathic effects is a promising means to control Parthenium. Two approaches are generally taken, i.e., (1) maintaining natural biodiversity and (2) selecting plants in target areas through planting [98]. The second approach involves the intentional manipulation of natural enemies to control harmful weeds, and there are some natural enemies of

Parthenium, which may be employed, such as *Cassia sericea, Cassia tora, Cassia auriculata,* and *Mirabilis jalapa.*

### 6.2. Physical Control

Hand hoeing before flowering and harvesting is the most cost-effective method of minimizing Parthenium. However, this method can cause health problems, such as allergic reactions. Some landowners have had success plowing the Parthenium weed when it is still in the tassel stage, which should be followed by sowing seed or planting the field. Hand weeding is a time-consuming and challenging method [99]. Burning is another strategy for managing Parthenium, but it requires a large amount of fuel [25,99].

### 6.3. Biocontrol Agents

Biocontrol techniques to reduce Parthenium constitute environmentally responsible and economically viable methods of control. The leaf rust fungus, *Puccinia xanthii* Schwein. var. *parthenii-hysterophorae*, and a stem-boring weevil, *Listronotus setosipennis* (Hustache), were found to be potential biocontrol agents for this weed [100]. Two biocontrol agents have been reported to have some effect in Ethiopia: the stem-galling moth and the seed-feeding weevil [101]. The stem-galling moth *Epiblema strenuana* reduced weed density, plant height, and flower production by 90%, 40%, and 82%, respectively. Furthermore, insects such as *Listronotus setosipennis* (stem-boring weevil), *Bucculatrix parthenica* (leaf-mining moth), and *Smicronyx lutulentus* (seed-feeding weevil) have demonstrated usefulness in Parthenium control [101,102]. After employing biocontrol measures since the mid-1990s, the infested region in Central Queensland has decreased [94]. The insects caused a large decline of Parthenium in confined areas, and the insects mainly feed on the weed's leaves, which ultimately damages the weed's ability to regenerate. Seeds and roots, on the other hand, maintain their normal biomass and are the primary drivers of the Parthenium's life cycle. The radicle and hypocotyl lengths of Parthenium seedlings exposed to cultural filtrates of *Fusarium oxysporum, Drechslera australiensis, Fusarium solani,* and *Drechslera hawaii ensis* were significantly reduced [54]. Great efforts have been made to regulate populations of Parthenium via various biocontrol agents, such as microbial pathogens, insects, and botanicals, over the last three to four decades [103]. The use of plant pathogens is an environmentally friendly solution that can be used in agro-ecosystems [104]. Spore suspension of *Cladosporium* sp. (MCPL 461) with 3% sucrose showed a 70%–80% germination reduction (*Lantana camara* and *Chromolaena odorata* were also tested) [105]. Recently, Kaur and Aggarwal [106] discovered that *Trichoconiella padwickii* triggered premature defoliation of Parthenium leaves, suggesting that it may be a potent regulator for this weed.

### 6.4. Use of Synthetic Herbicides

In locations where Parthenium's natural enemies are absent, chemical control is an effective technique of control. Glyphosate is a herbicide used the world over to control Parthenium [107]. Glyphosate offered greater than 93% control of Parthenium at the rosette stage, according to Javaid [108]. Javaid [108] also demonstrated that glyphosate, glufosinate, and trifloxysulfuron reduced the presence of Parthenium weeds by 86–95% at the bolted stage. Singh et al. [109] found that 2,4-D, atrazine, metribuzin, metsulfuron, chlorimuron, and glufosinate failed to suppress Parthenium at 18 WAT; however, glyphosate at 2.7 and 5.4 kg ha$^{-1}$ provided greater than 95% control of bolted plants. The combination of pendimethalin, bispyribac-sodium, and bensulfuron-methyl afforded excellent control of Parthenium (about 90%) [110]. Similarly, Parthenium plants can be killed by glyphosate and isoproturon [111,112]. Pre-emergence herbicides such as metolachlor plus atrazine, pendimethalin, and cyanazine plus atrazine have been shown to reduce the weed density of Parthenium [113]. Kaur et al. [37] observed that 15 days after spraying, and application of both 2,4-D EE (0.2%) and metribuzin (0.25% and 0.50%) was effective at controlling Parthenium: total eradication of the Parthenium population was achieved, and weed emergence was prevented. Bromoxynil+MCPA was discovered to be the most efficient,

as all of the herbicide doses totally killed the weed within 7 days at both developmental stages [114].

*6.5. Controlling by Use*

One of the most effective means of controlling this hazardous vegetation is management by use. There are numerous applications for Parthenium as a raw material or as an addition, some of which relate to herbicides, pesticides, insecticides, ethanol production, composting, green manure, nanoparticle synthesis, silkworm feed additives, and dicolorizing agents [25,115].

## 7. Chemical Constituents of *P. hysterophorus* and their Mechanisms

Allelochemicals produced by *P. hysterophorus* and some of the physiological effects are explained in Table 2 [16]. Allelochemicals have a range of physiological effects, including reduction in plant growth, excessive water absorption, leaf area expansion, and mineral nutrient absorption [116]. The mechanism of any phytotoxic mix involves making the cell membrane permeable in general; ion flux and hydraulic conductivity are thereby disrupted. The changes to the membrane can have an impact on the plant–water relationship, photosynthesis rate, oxygenation, ion equilibrium, and stomata opening and closing [16]. In the lettuce plant, the relative water content and leaf water content can be reduced by 10%. This is caused by a reduction in $CO_2$ supply to mesophyll cells and a reduction in photosynthesis. Gulzar and Siddiqui [117] reported that p-coumaric acid, ferulic acid, salicylic acid, and caffeic acids produce water stress in treated seedlings of *Glycine max* and *Sorghum bicolour*. According to Einhellig et al. [16], several allelochemicals reduced radicle dry weight and moisture content in sprouting mustard seedlings. Caffeine's impact on cell division is limited because it inhibits cell membrane growth and the synthesis of nucleic acid chains. The phenolic acids p-hydroxy phenylacetic acid, p-coumaric acid, and ferulic acid can affect the photosynthesis of plants [118].

**Table 2.** Different chemical constituents of *Parthenium hysterophorus*.

| Chemical Constituents | Mechanisms | Reference |
|---|---|---|
| Hydroxyphenyl acetic acid, Ferulic, and **p**-coumaric acid | Degradation of chlorophyll | [119] |
| **P**-hydroxybenzoic acid | Impedes seedling growth, generates water stress, and triggers stomatal closing | [120] |
| Caffeine | Prevents cell division, and abnormal root growth | [121] |
| Caffeic acid | Inhibits seed germination, plant development, water relationship, decrease chlorophyll contents | [122] |
| Caffeic, **p**-coumaric, ferulic, salicylic acids | Induces water stress, reduce hydraulic conductivity | [108,123,124] |
| Cinnamic acid and Benzoic acid | Damage to the thylakoid membrane disrupts or changes membrane permeability, causes ion efflux, and reduces chlorophyll content | [125] |
| Phenolic compounds | Reduction in hydraulic conductivity, net nutrient uptake | [123,125] |

## 8. Allelopathic Composites

A large number of allelochemicals can be separated from Parthenium plants to produce new compounds. Parthenium is a potent allelopathic crop; P-hydroxybenzoic acid and phenolic compounds being the important allelochemicals in its plants, and these allelochemicals are used for weed control [126]. The toxin's major components are "Parthenin" and other phytotoxic compounds, for example, caffeic acid, vanillic acid, anisic acid, chlorogenic acid, ferulic acid, fumaric acid, coumaric acid, quercelagetin, and hydroxybenzoic acid [10]. Much available research also highlights its effect on agriculture and natural habitats regarding health risks [127].

## 9. Phytochemistry

According to Roy et al. [128], Parthenium's chemistry is currently extensively understood. The elements of *P. hysterophorus* vary depending on their chemotype and geographical distribution of seeds. Plants that produce large amounts of bioactive compounds and medicinal plants can have in their leaves and flowers more than 45 sesquiterpene lactones[128]. *P. hysterophorus* has 23 chemicals that account for 90.1% or more of the volatile oils [129]. Phytochemicals are also categorized as either primary or secondary components, based on their roles in biological processes. Sugars, amino acids, vitamins, nucleic acid purines and pyrimidines, chlorophylls, and other primary constituents are among them. The remaining plant compounds, such as alkaloids, terpenes, flavonoids, lignans, plant steroids, curcumins, saponins, phenolics, and glucosides, are considered secondary compounds [24,121,122]. Rahmat et al. [11] have been investigating the mineral composition of *P. hysterophorus* and showed that significant amounts of potassium, calcium, magnesium, sodium, iron, zinc, copper, molybdenum, lead, lithium, nickel, cadmium, chromium, and manganese are present in Parthenium plants. Parthenin, hymenin, coronopilin, dihydroisoparthenin, hysterin, hysterophorin, and tetraneurin-A were found in the sesquiterpene lactones of *P. hysterophorus* from various geographical zones [10].

The phenolic acids isolated from *P. hysterophorus* plant parts extracted in different organic solvents includes: caffeic acid ($C_9H_8O_4$), p-coumaric acid ($C_9H_8O_3$), p-anisic acid ($C_8H_8O_3$), ferulic acid ($C_4H_4O_4$), fumaric acid ($C_4H_4O_4$), p-hydroxy benzoic acid ($C_7H_6O_3$), chlorogenic acid ($C_{16}H_{18}O_9$), neochlorogenic acid ($C_{16}H_{18}O_9$), protocatechuic acid ($C_7H_6O_4$), aerulic acid, and vanillic acid ($C_4H_4O_4$) [25]. Among flavovoids, quercetagetin-3,7-dimethyl ether, apigenin, kaempferol-3-o-glucoside, quercetin-3-o-glucoside, kaempferol-3-o-glucoarabinoside, luteolin, lignin, jaceidin, syringaresinol, santin, chrysoeriol, kaempferolglucoside, centaureidin, 6-hydroxykaempferol-3,6-dimethylether, tanetin (6-hydroxykaempferol-3,6,4-trimethylether), quercetinglucoside, 6-hydroxykempferol-3,7-dimethylether and kaempferol-glucoarabinoside can be found [128,130,131]. The major allergen in *P. hysterophorus* pollen was identified by Gupta et al. [132] as a new hydroxyproline-rich glycoprotein. Das et al. [133] identified four acetylated pseudoguaianolides, as well as several other recognized compounds, from the flowers of *P. hysterophorus.* Venkataiah et al. [134] isolated a new sesquiterpenoid, charminarone, the first seco-pseudoguaianolide, from the complete plant, along with numerous other recognized chemicals. From the chloroform extract of this weed, Chhabra et al. [135] found three ambrosanolides.

The extract of different parts of Parthenium plant contains various pseudoguaianolide as: parthenin, anhydroparthenin, 2,3-dihydro-10α-hydroxyparthenin, 8β-hydroxy parthenin, ambrosin, 10α-hydroxyparthenin, tetraneurin-A, tetraneurin-E, hymenin, 15-deacetyltetraneurin-A, damsin, 2β-hydroxycoronopilin, 8β-hydroxycoronopilin, hysterin, scopoletin, conchasin-A, 3β-acetoxyneoambrosine, ambrosanolides, 8-β-acetoxyhysterone-C, deacetyltetraneurin-A, hysterone A-E, 8-β-acetoxyhysterone-C, dihydroxyparthenin, acetylated pseudoguaianolide, coronopilin, 13-hydroxyparthenin, charminarone, dihydroisoparthenin, 13-methoxydihydroparthenin, balchanin, costunolide, 2 β,13α-methoxydihydroparthenin, epoxyartemorin, 8-α-hydroxyestaiatin, 11,13-dihydroparthenin, 13-methoxy-11,13-d ihydroambrosin, 3-β-hydroxycostunolide, 5-β-hydroxyreynosin 13-methoxydihydroambrosin, 1-β-hydroxyarbusculin, and 13-methoxy-11,13-dihydro parthenin [136,137].

## 10. Conclusions

*P. hysterophorus* is one of the most common weeds in many areas throughout the world. The Parthenium plant is best known for its effects on natural ecology and its negative impacts on human and animal health. Thus far, various techniques have been used to control this toxic weed, such as mechanical, chemical, and biological control, but individually, these techniques have failed to stop *P. hysterophorus* proliferation. To combat the spread of this weed, multi-pronged tactics are necessary. To address this problem,

public awareness has to be developed, and a participatory approach to control the invasive weeds should be adopted. There is a need to encourage research on the use potential of this weed and to evaluate its efficacy in field trials. One of the most promising methods to control the weed is through proper use. It can be achieved through joint efforts of researchers, farmers, governmental and non-governmental agencies. The discovery of this weed's applications may also pave the path for the weed's indirect eradication. At present, although *P. hysterophorus* is considered a weed, its new uses are coming to the forefront. Parthenium can be used as an herbicide, pesticide, insecticide, raw material, or additive in a variety of industries, including paper, pulp, and dye industries, to name a few. Various studies have also revealed that Parthenium has potent antioxidant, antimicrobial, and anticancer properties. Its nutritive contents make it a potential composting agent, but more extensive research is needed to investigate it as a source of compost and a natural pesticide for various crops. There is a need to develop a low-cost, simple method for removing harmful allelopathic chemicals in order to exploit Parthenium in a useful way. Parthenium is an interesting weed due to having both harmful and beneficial effects in relation to crops, humans, and livestock. Furthermore, *P. hysterophorus* has many phenolic derivatives that are responsible for weed suppression. These phenolic derivatives should be investigated for their bioherbicide potential.

**Author Contributions:** H.M.K.B. wrote the first draft and incorporated the input from the reviews. A.S.J., M.S.A.-H., M.K.U., N.A., M.P.A. and F.R. reviewed the draft and improved the manuscript. All authors have read and agreed to the published version of the manuscript.

**Funding:** The authors are thankful to the Bangladesh Agricultural Research Council (BARC), Bangladesh, for adequate funding and other support through the Project NATP Phase-II, and to Universiti Putra Malaysia.

**Institutional Review Board Statement:** Not applicable.

**Informed Consent Statement:** Not applicable.

**Data Availability Statement:** Not applicable.

**Acknowledgments:** Many thanks go to the Ministry of Agriculture (MoA) of the People's Republic of Bangladesh, the Bangladesh Agricultural Research Council (NATP Phase-II Project, BARC), and the Bangladesh Agricultural Research Institute (BARI) for providing financial support, and grateful to the research project entitled " Efficacy of glufosinate-ammonium to control rice field weeds" (vote number- 6365400), "Parthenium hysterophorus: Biology, ecology, and sustainable management" (GP-IPB/2017/9523400) and the Universiti Putra Malaysia (UPM) for assistance.

**Conflicts of Interest:** The authors declare no conflict of interest.

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
