# Peer review of "A Mystic Weed, Parthenium hysterophorus: Threats, Potentials and Management"

_agronomy, doi:10.3390/agronomy11081514_

Round 1

Reviewer 1 Report

The reviewed work is of an overview character. The Authors collected a lot of information about an interesting plant, which they called the mystic weed. The presented data fully characterize this plant, its biology, threats to humans and animals, medicinal, chemical and allelopathic properties. Despite the many advantages and possibilities of using this plant, it is still a nuisance and easily spreading weed. In this respect, I feel insufficient for information on how to control it, especially with the use of chemical methods. If the Authors extend this fragment of the work by providing more examples and research, the work will be acceptable for publication. The manuscript is interesting, its content is well thought out and well configured in the subchapters.

Author Response

Author response

Manuscript title: Mystic weed, Parthenium hysterophorus: threats, potentials and management

Manuscript ID: (Agronomy-1268568)

Round 1

Reviewer 1 Report

Major comments

The reviewed work is of an overview character. The Authors collected a lot of information about an interesting plant, which they called the mystic weed. The presented data fully characterize this plant, its biology, threats to humans and animals, medicinal, chemical and allelopathic properties. Despite the many advantages and possibilities of using this plant, it is still a nuisance and easily spreading weed. In this respect, I feel insufficient for information on how to control it, especially with the use of chemical methods. If the Authors extend this fragment of the work by providing more examples and research, the work will be acceptable for publication. The manuscript is interesting, its content is well thought out and well configured in the subchapters.

Response to Reviewer 1 comments

Response: Thank you very much for your valuable comments and suggestions. Authors are grateful to you for great and constructive criticism of the manuscript. The manuscript is revised by giving emphasis on all the valuable suggestions and comments especially chemical methods with providing more example and research.

Reviewer 2 Report

The manuscript about mystic weed Parthenium hysterophorus is interesting and important because everyone in the world needs to know about this weed. This plant has allelopathic properties, is not only harmful to world agriculture, humans and livestock, but also it is used as an anti-oxidant, anti-cancer, anti-tumour and its solution is often used as a pesticide for various disease control, it is often used as an organic fertilizer because the quantity of N, Mg, Ca, K are high in this plant. However, the manuscript is of low quality, written irresponsibly:

  • The contains of the text does not match the section title – 4.2 and 4.4 section;
  • The contains of the figure does not match the figure title – figure 5;
  • “Isothiocyanates are a form of isothiocyanate…”- line 425. What did you want to say?
  • The numbering of the sections is inconsistent.

I have indicated other inaccuracies in the manuscript.

Round 1

Reviewer 2 Report

Major comments

The manuscript about mystic weed Parthenium hysterophorus is interesting and important because everyone in the world needs to know about this weed. This plant has allelopathic properties, is not only harmful to world agriculture, humans and livestock, but also it is used as an anti-oxidant, anti-cancer, anti-tumour and its solution is often used as a pesticide for various disease control, it is often used as an organic fertilizer because the quantity of N, Mg, Ca, K are high in this plant. However, the manuscript is of low quality, written irresponsibly:

  • The contains of the text does not match the section title – 4.2 and 4.4 section;
  • The contains of the figure does not match the figure title – figure 5;
  • “Isothiocyanates are a form of isothiocyanate…”- line 425. What did you want to say?
  • The numbering of the sections is inconsistent.

I have indicated other inaccuracies in the manuscript.

Response to Reviewer 2 comments

Response: Thank you very much for your valuable comments and suggestions. Authors are grateful to you for great and constructive criticism of the manuscript. The manuscript is revised by giving emphasis on all the valuable suggestions and comments. The point-by-point response of the reviewer comments are given bellow:

C1: The contains of the text does not match the section title – 4.2 and 4.4 section

Response to C1: Thank you for your advice. Section 4.2 and 4.4: According to the suggestions, author revised the section with latest information and deleted other imprecisions, and the presence information is matched with the Title.

C 2: The contains of the figure does not match the figure title – figure 5

Response to C2: Figure 5: Deleted some word and changed the figure title with the matching of contains as well.

C 3: “Isothiocyanates are a form of isothiocyanate…”- line 425. What did you want to say?

Response to C3: L425: Yes, thanks. This will be “Isothiocyanates are biologically active, inhibiting the germination and growth of exposed plant species”. Now the information is well organized and revised.

C 4: The numbering of the sections is inconsistent

Response to C4: Yes, the numbering of the sections checked properly and organized it.

Response to C2: Thanks. I have already revised and added some missing information in line 242 as per your comments.

Comments C3: In line 249, the authors can give better and clearer information by explaining in what sense the hexane concentration impacts the germination and on what organisms it does.

Response to C3: L249: In line 249, the effect of hexane extraction section 5.3. Already revised with clear information about hexane extraction and its impacts on crops growth.

Comments C4: Along the whole manuscript we can find sentences that are not properly build and consequently, the meaning is unclear. As an example, check line 165, meaning of “leaf extraction was not rigid” in line 166, line 284, use of “However” in line 292, line 346, the expression “allelochemical growth” in line 486 which is not correct, etc.). I recommend a careful reading of the whole manuscript and thoughtful correction of these mistakes.     

Response to C4: Thanks for your comments. We revised as per your comments and updated the whole manuscript carefully. Especially, improved line with relevant information of the above-mentioned line: 165, 166, 284, 292, 346 and 486 accordingly.

Reviewer 3 Report

This is a review paper about Parthenium hysterophorus, a species that is considered a weed or a beneficial plant, depending on the place and time and if it is used or not by humans in order to obtain a profit in different aspects such as a medicinal plant, phyto-remedial mediator, etc.

The authors have done an extensive literature research and they wrote a quite complete report on what is known about this plant, which it is now expanding in South East Asia. The manuscript is divided in two parts, the first one deals on the harmful effect of this plant and the second one on the beneficial effects.

As a whole, the manuscript is sound, with many different valuable information about this species brought together. After a careful reading of the manuscript, my only main concern is that part of the information given does not seem to be related to the weed. For example, all paragraphs dealing about allelopathy are a mere description of this phenomenon, its functioning and its application but with any reference to the weed (from line 332 to line 438 al least). Also, the phytochemistry paragraph (from line 462 on) apparently does not have any direct relation with the weed neither. I agree that the information given is very interesting and valuable to understand these phenomena but the paper is about Parthenium hysterophorus and not about the description of these processes. Some literature linking these processes and the weed should be given.

According to the format of the journal, any author name should not be cited in the text. Check lines 154 and 234. And I miss some references in lines 43 to 48.

Revise  … growth and [60]… in line242. It seems that something is missing.

In line 249, the authors can give better and clearer information by explaining in what sense the hexane concentration impacts the germination and on what organisms it does.

Along the whole manuscript we can find sentences that are not properly build and consequently, the meaning is unclear. As an example, check line 165, meaning of “leaf extraction was not rigid” in line 166, line 284, use of “However” in line 292, line 346, the expression “allelochemical growth” in line 486 which is not correct, etc.). I recommend a careful reading of the whole manuscript and thoughtful correction of these mistakes.     

In overall, I think is a very good paper, original and novel. After revising the grammar and the points raised above about the content, I think it can be suitable for publication.

Round 1

Reviewer 4 Report

Major comments

This is a review paper about Parthenium hysterophorus, a species that is considered a weed or a beneficial plant, depending on the place and time and if it is used or not by humans in order to obtain a profit in different aspects such as a medicinal plant, Phyto-remedial mediator, etc.

The authors have done an extensive literature research and they wrote a quite complete report on what is known about this plant, which it is now expanding in South East Asia. The manuscript is divided in two parts, the first one deals on the harmful effect of this plant and the second one on the beneficial effects.

As a whole, the manuscript is sound, with many different valuable information about this species brought together. After a careful reading of the manuscript, my only main concern is that part of the information given does not seem to be related to the weed. For example, all paragraphs dealing about allelopathy are a mere description of this phenomenon, its functioning and its application but with any reference to the weed (from line 332 to line 438 al least). Also, the phytochemistry paragraph (from line 462 on) apparently does not have any direct relation with the weed neither. I agree that the information given is very interesting and valuable to understand these phenomena but the paper is about Parthenium hysterophorus and not about the description of these processes. Some literature linking these processes and the weed should be given.

According to the format of the journal, any author name should not be cited in the text. Check lines 154 and 234. And I miss some references in lines 43 to 48.

Revise…. growth and [60]… in line242. It seems that something is missing.

In line 249, the authors can give better and clearer information by explaining in what sense the hexane concentration impacts the germination and on what organisms it does.

Along the whole manuscript we can find sentences that are not properly build and consequently, the meaning is unclear. As an example, check line 165, meaning of “leaf extraction was not rigid” in line 166, line 284, use of “However” in line 292, line 346, the expression “allelochemical growth” in line 486 which is not correct, etc.). I recommend a careful reading of the whole manuscript and thoughtful correction of these mistakes.     

In overall, I think is a very good paper, original and novel. After revising the grammar and the points raised above about the content, I think it can be suitable for publication.

Response to Reviewer 4 comments

Response: Thank you very much for your valuable comments and suggestions. Authors are grateful to you for great and constructive criticism of the manuscript. The manuscript is revised by giving emphasis on all the valuable suggestions and comments with edited properly. 

I agreed with your comments from, line 332 to line 438: All necessary related information and references added accordingly.

L462: In the section phytochemistry, all irrelevant information to weed was discarded and also added some information that’s are relevant to weed properly.

The point-by-point response of the reviewer comments are given bellow:

Comments C1: According to the format of the journal, any author name should not be cited in the text. Check lines 154 and 234. And I miss some references in lines 43 to 48.

Response to C1: L154-234 and L43-48: Thanks. I already checked and revised it according to the format of the journal.

Comments C2: Revise…. growth and [60]… in line242. It seems that something is missing.

Response to C2: Thanks. I have already revised and added some missing information in line 242 as per your comments.

Comments C3: In line 249, the authors can give better and clearer information by explaining in what sense the hexane concentration impacts the germination and on what organisms it does.

Response to C3: L249: In line 249, the effect of hexane extraction section 5.3. Already revised with clear information about hexane extraction and its impacts on crops growth.

Comments C4: Along the whole manuscript we can find sentences that are not properly build and consequently, the meaning is unclear. As an example, check line 165, meaning of “leaf extraction was not rigid” in line 166, line 284, use of “However” in line 292, line 346, the expression “allelochemical growth” in line 486 which is not correct, etc.). I recommend a careful reading of the whole manuscript and thoughtful correction of these mistakes.     

Response to C4: Thanks for your comments. We revised as per your comments and updated the whole manuscript carefully. Especially, improved line with relevant information of the above-mentioned line: 165, 166, 284, 292, 346 and 486 accordingly.

Reviewer 4 Report

The paper is interesting and provides a good review of Parthenium.  The paper needs extensive editing to improve readability and reduce redundancy.  All extraneous information needs to be removed.  Table 2 and table 3 should be deleted.  Most of the section 6.5 Allelopathic shelter crops is extraneous and should be removed.

The following sections need to be rewritten to im6prove clarity:

L 34-40; L. 47; 61; 127; 147-157; 159-169; 171: (Parthenium's active concepts); 182-184; 188-190; 200-201;

204: you have describe numerous ways to use Parthenium, and then you state: "Parthenium has sevearl negative properties and isnt used in any way." please explain the contradiction.

L. 214; 248-251; 274-276;

L 321-330: mostly not relevant to Parthenium.

L. 405-415: mostly not relevant to Parthenium

482-483: unclear

484-495: clarify positives and negatives for the conclusion.

Round 1

Reviewer 3 Report

 Major comments

The paper is interesting and provides a good review of Parthenium.  The paper needs extensive editing to improve readability and reduce redundancy.  All extraneous information needs to be removed.  Table 2 and table 3 should be deleted.  Most of the section 6.5 Allelopathic shelter crops is extraneous and should be removed.

The following sections need to be rewritten to improve clarity:

L 34-40; L. 47; 61; 127; 147-157; 159-169; 171: (Parthenium's active concepts); 182-184; 188-190; 200-201;

204: you have described numerous ways to use Parthenium, and then you state: "Parthenium has sevearl negative properties and isnt used in any way." please explain the contradiction.

  1. 214; 248-251; 274-276;

L 321-330: mostly not relevant to Parthenium.

  1. 405-415: mostly not relevant to Parthenium

482-483: unclear

484-495: clarify positives and negatives for the conclusion.

Response to Reviewer 3 comments

Response: Thank you very much for your valuable comments and suggestions. Authors are grateful to you for great and constructive criticism of the manuscript. The manuscript is revised by giving emphasis on all the valuable suggestions and comments with edited and reduced redundancy properly.  All extraneous information has been removed. 

With proper respect to you, here I state that Table 3 deleted but table 2 still have.  Because table 2 shows the evidence of Allelochemicals produce from P. hysterophorus and some of the physiological effects from its are shown at a glance in Table 2. Another changed the hints: Chemical constituents and their mechanisms instead of Mode of action. I hope it will help to make the manuscript resourceful, informative and readers seen at a glance those who are not known as well as increase the attractiveness of the manuscript.

However, if you suggest me or assume that there is no significant important in the table 2, the I will discard this. There is no objection from my sight to delete table 2.

Section 6.5: Yes, from the section 6.5 thoroughly read and removed the extraneous information with revised. The point-by-point response of the reviewer comments are given bellow:

Comments C1: L 34-40; L. 47; 61; 127; 147-157; 159-169; 171: (Parthenium's active concepts); 182-184; 188-190; 200-201;

Response to C1: Thank you very much. Yes, I read thoroughly according to the mentioned line from 34 to 201. All are revised and added latest information as per your comments and suggestions.

Comments C2: 204: you have described numerous ways to use Parthenium, and then you state: "Parthenium has several negative properties and isn’t used in any way." please explain the contradiction.

Response to C2: L204: Yes, some information was missing. I already rearranged and added latest information in the line 204 on the basis of your comments.

Justification: With respect to you for your information. Actually, Parthenium plant is best known for its effects on natural ecology as well as its negative impact on human and animal health. Parthenium plant can be used as an herbicide, pesticide, insecticide, and as a raw material or additive in a variety of industries, including paper, pulp, and dye industries, to name a few. Various studies have also revealed that Parthenium plant have potent antioxidant, antimicrobial, and anticancer properties. Its nutritive contents make it a potential compost, but more extensive research is needed to investigate it as compost and natural pesticide for various crops. It has beneficial and harmful sights also and called as mystic weed.

Comments C3: L. 214; 248-251; 274-276;

Response to C3: Yes, according to your comments all above mentioned line read carefully and revised it properly.

Comments C4: L 321-330: mostly not relevant to Parthenium.

Response to C4: L321-330: Yes, according to your comments not relevant to Parthenium line removed properly and the relevant with latest information was added.

Comments C5: L. 405-415: mostly not relevant to Parthenium

Response to C5: L405-415: Yes, not relevant to Parthenium line removed properly and some relevant information was added accordingly.

Comments C6: 482-483: unclear

Response to C6: L482-483: Thanks for your comments. I deleted some unclear line and added some information properly.

Comments C7: 484-495: clarify positives and negatives for the conclusion

Response to C7: L484-495: Yes, according to your comments, already modified and clarified of positive and negatives sight in the conclusion section.

Round 2

Reviewer 3 Report

The authors have divided the paper into nine sections:

  1. Introduction
  2. Botanical description
  3. Harmful effects
  4. Beneficial effects
  5. Herbicidal effects of different solvents from the plant
  6. Controlling Parthenium
  7. Allelopathy
  8. Phytochemistry
  9. Economic benefits.
  10.  

My main issue with this paper is its structure as I already pointed out in my first revision. I got again confused when reading some subsections from section “7. Allelopathy” to 9 because some paragraphs are not apparently related to the weed and others seem that they should be placed in section 4 (or even 3). For example:

  1. In Section 7.4 the authors talk for first time about the “Berseem’s allelopatic behaviour”. What is this behaviour and what relationship has it with Parthenium? If “Berseem” is an allelopathic compound that comes from Partenium, I wonder why this information is not given in section “4. Beneficial effects”.
  2. Section 7.5 talks about the allelopathic effect of Brassica exudates. What is the relationship with Parthenium? Is this a control measure for this weed? If yes, consider to include it in section 6.
  3. Section 8, which is very short and with some sentences requiring a revision (see next comments below), seems that it should appear in section 3 or 4.
  4. The economic benefits reported in section 9 are, actually, beneficial effects. Consider to include in section 4

Alternatively, if this structure of ten points wants to be kept, the authors should make a clearer distinction between the content of the sections by, perhaps, with subsections and/or giving a more distinctive title to each section.

Moreover, there are quite a few typo mistakes that should be corrected throughout the manuscript:

  1. Too many spaces between words (L38, 162, 168, 169, 170,198, 246, 301, 346, 372, 380, 570 so far) or none (L292, 309, 432, 480).
  2. Repeated dots (i.e.: 159, 564, 566).
  3. Please, delete the comma after the author name and before the citation number and dots before the citation number (L282, 305, 311, 382, 387, 431, 459, 474, 549
  4. hysterophorus and, in general, all Latin names of plants and animals should be written in italics (L33, 172, 173, 197, 209, Table 1, 223, 265, 274, 275, 277, 278, 310, 311, 385).
  5. Revise alignment of title of section 3 and 3.2
  6. The meaning of acronyms should be done the first time they appear. What does the acronym RBC mean in L241?
  7. The sentence of the line 262 and 263 looks incomplete. The same for sentences in lines L472, 544-45 and 549-51.
  8. Delete the initials of the given names of the authors (L251, 252).
  9. L246: leaf, with lower case letter.
  10. Please, revise lines 282, 475.
  11. L284: when starting a new sentence with the name of an author, his name has been always given (i.e. L282, 466). Please, check this point in this sentence.
  12. Please, remove the frame of figure 9 and table 4.

The number of references in the Reference section (167) does not match with the number of references of the main text (147).

Reviewer 4 Report

Paper is much improved from edition 1.  There are still several grammatical errors and misspellings.  Some sentences are missing subjects or predicates. There are numerous improper choice of words, mainly verbs.  I suggest having a native English speaker review the paper before final submission.

Author Response

 Major comments

Paper is much improved from edition 1.  There are still several grammatical errors and misspellings.  Some sentences are missing subjects or predicates. There are numerous improper choice of words, mainly verbs.  I suggest having a native English speaker review the paper before final submission.

General Response: Thank you very much for your valuable comments and suggestions. Authors are grateful to you for great and constructive criticism of the manuscript. The manuscript is revised by giving emphasis on all the valuable suggestions and comments with edited by MDPI English editorial section.

This manuscript is a resubmission of an earlier submission. The following is a list of the peer review reports and author responses from that submission.